# Constructing a Knowledge Graph from Textual Descriptions of Software Vulnerabilities in the National Vulnerability Database

**Anders Mølmen Høst**
Simula Research Laboratory
& University of Oslo
Oslo, Norway
andersmh@simula.no

**Pierre Lison**
Norwegian Computing Center
& University of Oslo
Oslo, Norway
plison@nr.no

**Leon Moonen**
Simula Research Laboratory
& BI Norwegian Business School
Oslo, Norway
leon.moonen@computer.org

## Abstract

Knowledge graphs have shown promise for several cybersecurity tasks, such as vulnerability assessment and threat analysis. In this work, we present a new method for constructing a vulnerability knowledge graph from information in the National Vulnerability Database (NVD). Our approach combines named entity recognition (NER), relation extraction (RE), and entity prediction using a combination of neural models, heuristic rules, and knowledge graph embeddings. We demonstrate how our method helps to fix missing entities in knowledge graphs used for cybersecurity and evaluate the performance.

## 1 Introduction

An increasing number of services are moving to digital platforms. The software used on these digital platforms is, unfortunately, not without flaws. Some of these flaws can be categorized as security vulnerabilities that an attacker can exploit, potentially leading to financial damage or loss of sensitive data for the affected victims. The National Vulnerability Database (NVD)[1] is a database of known vulnerabilities which, as of January 2023, contains more than 200 000 vulnerability records. The Common Vulnerability and Exposures (CVE) program[2] catalogs publicly disclosed vulnerabilities with an ID number, vulnerability description, and links to advisories. NVD fetches the data from CVE and provides additional metadata such as weakness type (CWE) and products (CPE). CWEs are classes of vulnerabilities (CVEs), for example, CWE-862: Missing Authorization contains all CVEs related to users accessing resources without proper authorization. A CPE is a URI string specifying the product and its version, for example, *cpe:2.3:a:limesurvey:limesurvey:5.4.15* is the CPE for the survey app Limesurvey with version 5.4.15. Keeping the information in the database up to date is important to patch vulnerabilities in a timely manner. Unfortunately, patching becomes increasingly difficult as the yearly number of published vulnerabilities increases.[3]

To automatically extract relevant information from vulnerability descriptions, named entity recognition (NER) and relation extraction (RE) can be applied as shown in Fig. 1. The extracted information can be stored as triples in a knowledge graph (KG). As the extracted triples might be incorrect or missing, knowledge graph embeddings (KGE) can be used to learn the latent structures of the graph and predict missing entities or relations.

The work described in this paper is based on the master thesis by the first author. We investigate how NLP and KGs can be applied to vulnerability records to predict missing software entities. More specifically, we address the following research question: *RQ: Can our knowledge graph predict vulnerability weakness types and vulnerable products?* The contributions of this paper include: (1) An approach for extracting and assessing vulnerability data from NVD; (2) A vulnerability ontology for knowledge graph construction; (3) A rule-based relation extraction model.

## 2 Related Work

We distinguish the ensuing areas of related work:
**Labeling:** Labeled data may not always be available to train supervised learning models for tasks including NER and RE. To address this problem, distant supervision aims at proposing a set of labeling functions for the automatic labeling of data. Bridges et al. (2014) applied distant supervision using a cybersecurity corpus. Their ap-

---

[1] https://nvd.nist.gov/
[2] https://www.cve.org/
[3] https://nvd.nist.gov/general/nvd-dashboard

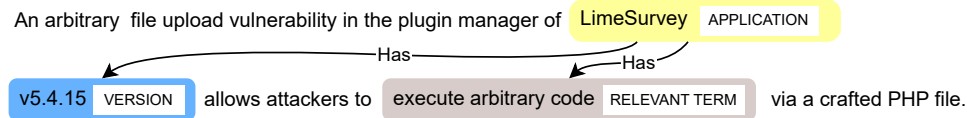

Figure 1: Example of a CVE with labels

proach includes database matching using the CPE vector, regular expressions to identify common phrases related to versioning, for example, "before 2.5", and *gazetteers*, which are dictionaries of vulnerability-relevant terms, such as "execute arbitrary code".

After manual validation of the labeled entities, Bridges et al. (2014) report a precision of 0.99 and a recall of 0.78.

**Named Entity Recognition:** Training NER models on labeled data are useful as distant supervision depends on assumptions about the input data, which does not always hold. For example, in the case of NVD, if the new data is missing CPE information. Machine learning models are not dependent on such metadata, and, as a consequence can generalize better to new situations. Bridges et al. (2014) propose NER based on the Averaged Perceptron (AP). The conventional perceptron updates its weights for every prediction, which can over-weight the final example. The averaged perception keeps a running weighted sum of the obtained feature weights through all training examples and iterations. The final weights are obtained by dividing the weighted sum by the number of iterations.

Gasmi et al. (2019) propose another NER model based on a long short-term memory (LSTM) architecture. The authors argue that it can be more useful when the data set has more variation, as the LSTM model does not require time-consuming feature engineering. However, their results show it is not able to reach the same level of performance as Bridges et al. (2014).

SecBERT[4] is a pre-trained encoder trained on a large corpus of cybersecurity texts. It is based on the BERT architecture (Devlin et al., 2019) and uses a vocabulary specialized for cybersecurity. SecBERT can be fine-tuned for specific tasks such as NER.

Another pre-trained encoder similar to SecBERT is SecureBERT, proposed by Aghaei et al. (2022). SecureBERT leverages a customized

---

[4]https://github.com/jackaduma/SecBERT

tokenizer and an approach to alter pre-trained weights. By altering pre-trained weights, Secure-BERT aims to increase understanding of cyber security texts while reducing the emphasis on general English.

**Relation Extraction:** Relations between named entities can be discovered with RE. Gasmi et al. (2019) propose three RE models for vulnerability descriptions from NVD based on LSTMs. Their best-performing model achieves a precision score of 0.92. For labeling the relations, Gasmi et al. (2019), applies distant supervision (Jones et al., 2015). Gasmi et al. (2019) does not manually evaluate their labels before using them in the LSTM models; however, the approach is based on Jones et al. (2015), which indicates 0.82 in precision score after manual validation. Both NER and RE are important components for constructing knowledge graphs from textual descriptions. We explore several knowledge graphs related to cybersecurity in the next section.

**Knowledge Graphs in Cybersecurity:**

CTI-KG proposed by Rastogi et al. (2023), is a cybersecurity knowledge graph for Cyber Threat Intelligence (CTI). CTI-KG is constructed primarily from threat reports provided by security organizations, describing how threat actors operate, who they target, and the tools they apply. Rastogi et al. (2023) manually labels a data set of approximately 3000 triples with named entities and relations. This labeled data is then used for training models for NER and RE for constructing the KG. CTI-KG also uses KGE to learn latent structures of the graph and predict incomplete information.

Here, Rastogi et al. (2023) applies TuckER, a tensor decomposition approach proposed by Balaževi´c et al. (2019), which can be employed for knowledge graph completion. TuckER can represent all relationship types (Balaževi´c et al., 2019), as opposed to earlier models. For example, TransE proposed by Bordes et al. (2013) has issues modeling 1-to-$n$, $n$-to-1, and $n$-to-$n$ relations (Lin et al., 2015). An example of a 1-to-$n$ relationship in a cybersecurity context is the relationship between

CVEs and CPEs. Whereas a CVE can have multiple CPEs, a CPE can only have one CVE.

As CTI-KG focuses on threats, another KG, VulKG (Qin and Chow, 2019), is constructed from vulnerability descriptions from NVD. VulKG consists of three components, a vulnerability ontology, NER for extracting entities from the vulnerability descriptions, and reasoning for discovering new weakness (CWE) chains. After extracting entities, relations between these can be found using the VulKG ontology (Qin and Chow, 2019). The final step of the framework presented by Qin and Chow (2019) is the reasoning component which is based on chain confidence for finding hidden relations in the graph.

Similarly to VulKG, we construct our KG from vulnerability descriptions in NVD. However, VulKG depends on training NER models from scratch, while we instead depend on a pre-trained model fine-tuned to our data. Contrary to training the model from scratch, the pre-training approach utilizes an existing model already trained on a large dataset. Consequently, fine-tuned models can learn patterns in the new data set more quickly.

## 3 Methods

Our approach is shown in Fig. 2 and gives an overview of the construction of the vulnerability knowledge graph from CVE records. We discuss the different steps below. For replication, we share details about the hyperparameter tuning of various models in the appendices.

**Data:** Our dataset is downloaded in JSON format from NVD, and the pipeline consists of multiple steps before predicting missing or incorrect labels as the final step. The data set consists of all CVE records from 2003 to 2022, which contains approximately 175 000 CVEs. The CVE records are labeled using the distant supervision approach proposed by Bridges et al. (2014).

**Named Entity Recognition:** We train two architectures: Averaged Perceptron and SecBERT.

*Averaged Perceptron (AP):* AP is a feature-engineered model, and we use the same features as Bridges et al. (2014) Due to computational constraints in the AP model, we restricted our training data to 4000 CVEs.

We first replicate their approach and separately trained and evaluated two AP models, one for IOB-labeling and one for domain-labeling, using the distant supervision-generated labels. In practice, when a new CVE is published, we only have access to the textual description. Since the IOB labels are input features to the domain model, those must be predicted first. Thus, in our second experiment, we again train two AP models, but use the predicted IOB labels as input to the domain labeling, instead of the generated labels.

*SecBERT:* In addition to AP, we use the pre-trained SecBERT model for NER. A significant difference from AP is that SecBERT jointly extracts IOB and domain labels. Moreover, as SecBERT is significantly faster than AP, there is no need to restrict the dataset. We split our data into 60/20/20 for training, evaluation, and testing.

**Relation Extraction:** For relation extracting, we use an *ontology* illustrated in Fig. 3, to guide their creation: When two entities of type $A$ and $B$ are detected in a CVE, a relation between the two is created if the ontology has an edge between types $A$ and $B$.

Note that entities are connected to their corresponding CVE-ID and CWE-ID, and we concatenate multi-word entities based on their IOB labels.

The vulnerability descriptions are generally written so that vendors are followed by their products which are then followed by their versions. Thus, we can derive relations between vendor, product, and version by looking at the word order. We also make relations from relevant terms to the corresponding CVE ID entity, and through the CVE-ID the relevant terms are connected to the corresponding vendors, products, and versions.

**Entity Prediction:** To answer the RQ, our KG should predict weakness types (CWEs) and products (CPEs). Given a head entity and a relation as input, the task of entity prediction is to find the tail entity, which is the final step of our KG. $Hits@n$ and *mean reciprocal rank* (MRR) are standard metrics used for entity prediction. For each input example, the embedding algorithm assigns a confidence score to all possible triples. These triples are then ranked by confidence scores, where the triple with the highest confidence is the most plausible to be true according to the model. The $Hits@n$ metric measures the number of times the true triple is ranked among the top $n$ triples. We use the processed triples from the RE model as input to our entity prediction model, where TuckER is the chosen architecture. The triples from our RE model are considered ground truth. TuckER removes the

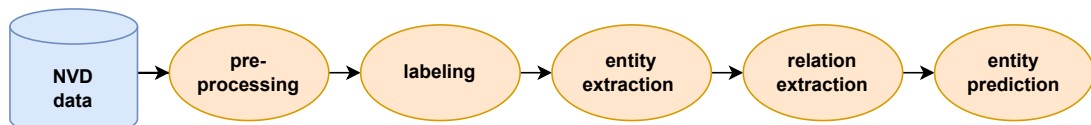

Figure 2: The figure illustrates the steps in our approach. We start by downloading our data from NVD, pre-processing the data, and adding labels to the entities. With our labeled data, we perform NER and RE to construct the KG. Because missing entities might occur in the KG, we predict these in the last step.

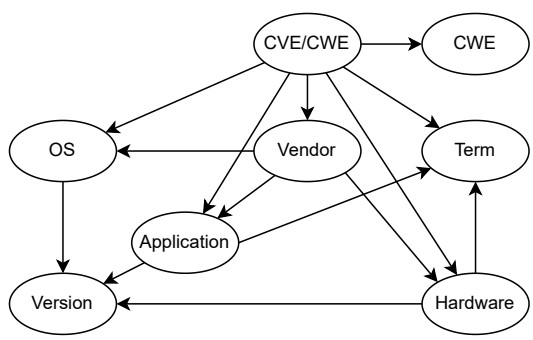

Figure 3: Ontology for relation extraction. The edges should be interpreted as, for example, "a vendor *has* a product", "a product *has* a version", "a CVE vulnerability *has* a CWE type"

tail entities from the ground truth before predicting these based on entity and relation embeddings. We perform data augmentation by reversing all the relational triples. The data set is split in 80/10/10 percent for training, validation, and testing. We select the best model by refining the four combinations proposed by Balažević et al. (2019) with an additional grid search.

## 4 Results and discussion

Our empirical evaluation uses the CVE dataset discussed in Section 3. For replication, the parameters of the best-performing models are in the appendices.

**NER:** NER results are presented in Table 1. We see that SecBERT outperforms AP on all metrics.

We compare our reproduction results with the results reported by Bridges et al. (2014) in Ta-

Table 1: NER evaluation results for the averaged perception and the fine-tuned SecBERT model.

| NER Model | Precision | Recall | $F_1$ |
|---|---|---|---|
| Averaged perceptron | 0.925 | 0.84 | 0.88 |
| Fine-tuned SecBERT | 0.93 | 0.93 | 0.93 |

Table 2: Our reproduction results compared to those reported by Bridges et al. (2014)

| Author | Labeling | Precision | Recall | $F_1$ |
|---|---|---|---|---|
| Høst et al. | IOB | 0.93 | 0.93 | 0.93 |
| | Domain | 0.94 | 0.94 | 0.94 |
| Bridges | IOB | 0.97 | 0.97 | 0.96 |
| | Domain | 0.99 | 0.99 | 0.99 |

ble. 2. Where Table 1 shows the performance with all labels in place, individual IOB and domain labeling performance are reported in Table 2. The AP model was based on Bridges et al. (2014), which implemented their experiments in OpenNLP and Python. We reused their Python code for our reproduction. Note that the results on our data are below the reports by Bridges et al.. The authors indicated that they experienced slightly better performance using OpenNLP, which *could* be the reason for the difference in score. Unfortunately, they do not provide any explanation of this difference or why it occurs. Contrary to Bridges et al. (2014), we are not interested in the performance of IOB and domain labeling measured individually. In our approach, the NER model should be used to extract entities from new data that can form triples in our KG. When a new CVE is published, we can access the textual description without any labels. Using Bridges' approach, we first need to use the IOB model, and then the predicted IOB labels can be used as input features to the domain model responsible for the final prediction.

To the best of our knowledge, we can not analytically combine the IOB model and domain model results reported by Bridges et al.. As such, we rely on our own experimental results, which show that the performance of the fine-tuned SecBERT model outperforms the AP model.

**Relation Extraction:** We did not have any ground truth data when evaluating our RE approach, as a consequence, we manually validated

Table 3: Performance metrics for our entity prediction model compared to Rastogi et al. (2023).

| Model | Hits@10 | Hits@3 | Hits@1 | MRR |
|-------|---------|--------|--------|-----|
| Høst et al. | 0.760 | 0.728 | 0.682 | 0.710 |
| Rastogi | 0.804 | 0.759 | 0.739 | 0.75 |

a sample of 100 extracted triples. From this sample, we measured a precision score of 0.77. While Jones et al. (2015) has proposed a semi-supervised approach for labeling relations, they focus on a broader data set than we do. We, therefore, choose to identify relations based on our proposed ontology in Fig. 3. Our RE approach could not reach the level of Jones et al. (2015), which reported 0.82 in precision score. For future work, one idea to improve RE is to utilize CPE vectors for relation labeling in addition to our proposed rules. Then we can train machine learning models on top of our labeled data using pre-trained variations of BERT models.

**Entity Prediction:** During the relation extraction, we extracted approximately two million triples. As we further reversed all triples, four million triples were used as input to the model.

In Table. 3, we compare our best-performing model with the results presented in Rastogi et al. (2023), which uses the same model architecture, TuckER, on threat reports. The input data are assumed to be true, and evaluation performance is not manually validated.

We choose TuckER as our embedding algorithm for entity prediction as it is the current state-of-the-art model measured on standard data sets (Balažević et al., 2019). The idea is that TuckER captures latent structures of our KG. TuckER encodes the input triples as vector embeddings based on encoded characteristics and can use these embeddings to predict missing entities. For example, if two CVEs share important characteristics such as vulnerability-relevant terms and affected products, then according to the theory, they should belong to the same neighborhood in a vector space. Consequently, TuckER could predict that the CVEs belong to the same CWE.

$Hits@n$ and *mean reciprocal rank* (MRR) are standard metrics used for entity prediction. Given a head entity and a relation, the task is to predict the tail entity. For each example, the embedding algorithm assigns a confidence score to all possible triples. These triples are then ranked by confidence scores, where the triple with the highest confidence is the most plausible to be true according to the model. The $Hits@n$ metric measures the number of times the true triple is ranked among the top $n$ triples.

As a benchmark to measure our performance, we use the results presented in Rastogi et al. (2023), which also uses TuckER for entity prediction. Rastogi et al. (2023) has reported a Hits@10 metric of 0.804, which is better than our reported results seen in Table 3. We believe that more precise and consistent input labels can be the reason for this, where a limitation of our approach is that we aim at predicting CVE-IDs which are unique for each vulnerability description. We consider the task of predicting CVE-IDs as less important for our model as these will always be attached to the CVE description from our raw data. Balažević et al. (2019) addresses that future work might incorporate background knowledge on relationship types. Avoiding predicting CVE-IDs is one example of such background knowledge.

Another reason for the difference could be that some CWEs overlap and share many of the same entities making it more difficult for our model to discriminate between CWEs.

## 5 Conclusion

This paper proposes a vulnerability knowledge graph constructed from textual CVE records from the National Vulnerability Database (NVD). The graph construction relies on a pipeline including NER, relation extraction, and an entity prediction model based on the TuckER framework.

As future improvements, we are interested in better labeling of relations through distant supervision approaches and the integration of BERT models for relation extraction.

## Acknowledgements

The research presented in this paper was supported by the Research Council of Norway through projects secureIT (grant #288787) and CLEANUP (grant #308904). The empirical evaluation used the Experimental Infrastructure for Exploration of Exascale Computing (eX3), supported by the Research Council of Norway through grant #270053.

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

# Appendices

## A   SecBERT NER Tuning

We first perform a grid search (four epochs) over the 20 parameter combinations recommended by the BERT authors.[5] The grid consisted of batch sizes: {8, 16, 32, 64, 128}, and learning rates: {3e-4, 1e-4, 5e-5, 3e-5}. The most promising candidates were then trained for ten epochs.

## B   Entity Prediction Tuning

For tuning hyperparameters, we follow two strategies: First, we train the same four combinations as was done by (Balažević et al., 2019). These four models were run for 100 epochs and based on the intermediate results, the most promising model was run for additional 200 epochs such that this model was trained for 300 epochs in total. We select the best-performing model and based on its characteristics set up an additional grid search covering 36 hyperparameter combinations on smaller subsets of the data. To avoid overfitting, two models were trained and evaluated for each of the hyperparameter combinations on different subsets. Our grid consisted of values of hidden dropouts: {0, 0.1, 0.2}, learning rates: {0.001, 0.01, 0.1} and dimensions: {10, 30, 200}. The parameters from the most promising candidate were used for training another model for 300 epochs on the full dataset.

## C   Best Model for NER

The best SecBERT model for NER was trained with a learning rate of 5e-5 and a batch size of 8.

## D   Best Model for Entity Prediction

The following are the hyperparameters of the best-performing TuckER model:

| Model | TuckER |
|---|---|
| num_iterations | 300 |
| edim | 200 |
| rdim | 30 |
| lr | 0.001 |
| input_dropout | 0.2 |
| hidden_dropout1 | 0.1 |
| hidden_dropout2 | 0 |
| batch_size | 128 |
| label_smoothing | 0.1 |
| dr | 1 |

---

[5] https://github.com/google-research/bert