# OpenReview forum: "Constructing a Knowledge Graph from Textual Descriptions of Software Vulnerabilities in the National Vulnerability Database"
_NoDaLiDa/2023/Conference — NoDaLiDa 2023_

### Official Review · Reviewer_eq2e · 2023-03-07
**Interesting paper on the use of knowledge graphs for cybersecurity language modeling. Improvements may require modifying the approach.**

**Rating:** 7
**Confidence:** 4

**Review:**

The authors present a paper in the area of cybersecurity. Their new method is based on constructing a vulnerability knowledge graph from information residing in the National Vulnerability Database (NVD). The approach combines NLP traditional tasks like named entity recognition (NER), relation extraction (RE), with entity prediction using a combination of neural models, heuristic rules and the key element which is made of knowledge graph embeddings. The average performance achieves 0.76 on Hits@10. As the authors affirm, this method may help fix missing entities in knowledge graphs. The paper addresses the following research question: RQ: Can our knowledge graph predict vulnerability weakness types and vulnerable products?
The key idea is trying to extract triples which can then be stored in a knowledge graph. Triples extracted may be incorrect or missing in some or more relevant elements: the knowledge graph could then be used to learn the latent structure and help predict missing elements of the triple, be it relation or entity.
The paper presents in turn a framework for extracting and assessing vulnerability data from the NVD; then the construction of a vulnerability ontology for knowledge graphs and finally a rule-based model for relation extraction.
The authors do NER by training two architectures: Average Perceptron and SecBERT, the latter being a pre-trained encoder trained on cybersecurity texts. It uses a vocabulary specialized for cybersecurity and can be fine-tuned for specific tasks.
In order to predict incomplete information a cybersecurity knowledge graph for Cyber Threat Intelligence (CTI) is built. CTI-KG is constructed mainly from threat reports provided by security organizations. CTI-KG uses KGE to learn latent structures of the graph and predict incomplete information by applying TuckER, a tensor decomposition framework which can be used for KG completion.
I fully agree with the conclusion of the authors when they affirm they need more precise and consistent input labels, and that a limitation of the approach is that they aim at predicting CVE-IDs which are unique for each vulnerability description. They also point out that problems arise due to CWEs overlapping and sharing many of the same entities making it more difficult for the model to discriminate between CWEs.
Also, as the authors of the paper on TuckER they cite, working on triples may suffer from the lack of sufficient background knowledge on individual relation properties which should be injected into the existing model.
Finally, knowledge graph embeddings are supervised learning models that learn vector representations of nodes and edges of labeled, directed multigraphs, and as such their predictions, even if redirected by a fine-tuning phase are always approximated whenever new text and new entities appear. Improving the performance obtained with SecBERT[1], which is certainly to be regarded state-of-the-art, may depend on the model or on the procedures used. In the latter case the approach suggested by SecureBERT maybe interesting. The authors developed a customized tokenizer as well as a method to alter pre-trained weights which obtained high F1 values. Their performance is remarkable. Increasing information is also what authors of [2] seem to suggest to boost performance and no tensor is used.

[1] Ehsan Aghaei1, Xi Niu1, Waseem Shadid1, and Ehab Al-Shaer, 2022. SecureBERT: A Domain-Specific Language Model for Cybersecurity, arXiv:2204.02685v3.
[2] Robert L. Logan IV, Nelson F. Liu, Matthew E. Peters, Matt Gardner, Sameer Singh, 2019. Barack’s Wife Hillary: Using Knowledge Graphs for Fact-Aware Language Modeling, in Proceedings of the 57th Annual Meeting of the Association for Computational Linguistics, pages 5962–5971.

**Paper Type:**

Long paper

---

### Official Review · Reviewer_HUu2 · 2023-03-08
**Review of NoDaLiDa submission "Constructing a Knowledge Graph from Textual Descriptions of Software Vulnerabilities in the National Vulnerability Database"**

**Rating:** 6
**Confidence:** 4

**Review:**

This paper describes a very specific use case for constructing a knowledge graph from a relational database using a fixed ontology. The results appear to be promising and potentially practically useful. The example given makes the reader want more of them :-). (This reviewer has deployed LimeSurvey on his corporate site...)

This paper will be of interest not primarily for its application scenario but more for the general case of using NER and RE tools on less structured databases. For this to be useful, the experiments would need to be replicated on another dataset in addition to the present one.

**Paper Type:**

Short paper

---

### Decision · Program_Chairs · 2023-03-17

Accept